# FEDERATED FEATURE TRANSFORMATION WITH SAMPLE-AWARE CALIBRATION AND LOCAL–GLOBAL SEQUENCE FUSION

## ABSTRACT

Tabular data plays a crucial role in numerous real-world decision-making applications, but extracting valuable insights often requires sophisticated feature transformations. These transformations mathematically transform raw data, significantly improving predictive performance. In practice, tabular datasets are frequently fragmented across multiple clients due to widespread data distribution, privacy restrictions, and data silos, making it challenging to derive unified and generalized insights. To address them, we propose a Federated Feature Transformation (FEDFT) framework that enables collaborative learning while preserving data privacy. In this framework, each local client independently computes feature transformation sequences and evaluates the corresponding model performances. Instead of exchanging sensitive original data, clients transmit these transformation sequences and performance metrics to a central global server. The server then compresses and encodes the aggregated knowledge into a unified embedding space, facilitating the identification of optimal feature transformation sequences. To ensure unbiased aggregation, we tackle three challenges in distributed tabular data: insufficient samples compromise statistics, limited feature diversity misses patterns, and sparse or correlated columns cause instability. We employ a sample-aware weighting strategy that favors clients with adequate size, rich diversity, and stable numerical properties. We also incorporate a server-side calibration mechanism to adaptively refine the unified embedding space, mitigating bias from outlier data distributions. Furthermore, to ensure optimal transformation sequences at both global and local scales, the globally optimal sequences are disseminated back to local clients. We subsequently develop a sequence fusion strategy that blends these globally optimal features with essential non-overlapping local transformations critical for local predictions. Extensive experiments are conducted to demonstrate the efficiency, effectiveness, and robustness of our framework. Code and data are publicly available.[1]

## 1 INTRODUCTION

Tabular data is ubiquitous across diverse domains and serves as a fundamental backbone for real-world decision-making applications. To facilitate insight extraction and enhance model performance, feature transformation systematically applies mathematical operations to raw features, producing more expressive and informative features. In real-world scenarios, tabular data is often distributed across multiple local clients, resulting in isolated data silos. Due to privacy and regulatory constraints, it becomes challenging to derive unified and generalizable insights from such fragmented sources. Overcoming these limitations imposed by data silos and achieving unified modeling of tabular data across diverse regions has emerged as a critical and active research direction. Existing works on feature transformation predominantly focus on centralized settings, assuming datasets reside on a single machine, thereby enabling these methods to utilize the complete feature space for effective refinement. They can be mainly classified into three categories: 1) Expansion reduction methods Horn et al. (2019); Khurana et al. (2016) generate numerous candidate features via mathe-

---

[1] https://anonymous.4open.science/r/FedFT_ICLR2026-3578

matical transformations, subsequently selecting the most informative subset; 2) Iterative feedback-based techniques: Khurana et al. (2018); Tran et al. (2016); Wang et al. (2022) progressively refine features using predictive performance feedback, typically optimized through reinforcement learning or evolutionary algorithms; 3) AutoML-driven frameworks Chen et al. (2019); Zhu et al. (2022); Wang et al. (2023) formalize feature transformation as neural architecture or policy search problems, aiming to identify optimal feature transformation sequences to enhance task-specific performance. But, due to centralized learning constraints, these methods are unable to aggregate knowledge from multiple clients, limiting their capability to derive unified and generalized feature insights.

To address these limitations, we propose a unified automated feature transformation framework capable of aggregating knowledge from diverse clients and deriving an optimal feature transformation sequence. Given the distributed nature of data across multiple clients and concerns about privacy, Federated Learning (FL) naturally emerges as a suitable paradigm, allowing collaborative learning without compromising local data confidentiality McMahan et al. (2017); Li et al. (2020); Karimireddy et al. (2020). However, adopting FL in this context introduces three major challenges:

- **Privacy-Preserving Globally Optimal Feature Transformation.** Learning globally optimal feature transformation sequences from distributed tabular data presents privacy challenges. In domains such as healthcare, tabular datasets often contain sensitive information, including treatment histories, personal demographics, and etc. Transmitting such data across clients introduces serious privacy risks. Thus, it is essential to develop mechanisms that support effective knowledge aggregation while strictly preserving data privacy.

- **Unbiased Aggregation under Non-IID and Imbalanced Distributions.** Client data exhibit heterogeneity in both feature distributions and sample sizes. Such disparity poses a unique challenge when aggregating feature transformation knowledge, as it can introduce bias into the global embedding space. This bias may lead to suboptimal transformation sequences and reduced feature space quality. Thus, it is critical to design aggregation strategies that explicitly account for data imbalance and distributional shifts across clients.

- **Effective Feedback from Global Server to Local Clients.** Ensuring that local clients benefit from global knowledge is crucial for sustainable collaboration. For instance, in financial applications, institutions may be reluctant to participate if they contribute data but receive no benefits. Thus, it is necessary to develop feedback mechanisms that adapt global transformation sequences to local contexts, promoting mutual benefit in this scenario.

To address these challenges, we propose a novel **FED**erated **F**eature **T**ransformation framework (**FEDFT**), which enables collaborative feature transformation across clients while preserving data privacy. The primary objective of FEDFT is to construct an optimized and generalizable feature space by aggregating transformation knowledge from heterogeneous local datasets without exposing raw data. Specifically, each client independently generates a collection of feature transformation records, where each record comprises a transformation sequence and associated performance, evaluated on the client's local tabular data. These records are then transmitted to a global central server for global knowledge aggregation. Notably, only the transformation sequences and corresponding predictive performance are shared, ensuring that sensitive raw data remains local and protected throughout the process. To effectively compress and leverage the collected transformation knowledge, we design an encoder-decoder-evaluator architecture. The encoder maps transformation sequences into a shared latent embedding space, the decoder reconstructs the sequences from their embeddings, and the evaluator estimates the expected model performance from the latent representation. Once the embedding space is constructed, we utilize gradient signals from the evaluator to guide exploration within the space, enabling the discovery of improved feature transformation sequences. To ensure unbiased aggregation throughout the optimization process, we introduce a sample-aware weighting mechanism that accounts for the varying reliability of transformation records across clients. Specifically, in tabular data contexts, we prioritize clients based on three factors: sample size for statistical validity, feature diversity for comprehensive pattern coverage, and numerical stability for robust metrics. This is complemented by a server-side calibration process to further mitigate bias from outlier distributions. To support both global generalization and local adaptability, the globally optimized transformation sequences are fed back to individual clients. We then develop a sequence fusion strategy that integrates these global sequences with critical, non-overlapping local transformations tailored to the specific predictive needs of each client. Finally, we conduct comprehensive experiments to validate the effectiveness, generalizability, and practical utility of the framework.

## 2 RELATED WORKS

**Automated Feature Transformation (AFT)** aims to refine or augment the original feature space so that machine learning models can more effectively capture complex, high-order relationships among variables. Most AutoML pipelines refine the input space in one of two ways: (i) by applying explicit statistical or arithmetic operators to create interpretable composite features Horn et al. (2019); Kanter & Veeramachaneni (2015); Khurana et al. (2016; 2018); Tran et al. (2016); Wang et al. (2022); Chen et al. (2019); Zhu et al. (2022); Wang et al. (2023), or (ii) by learning high-dimensional latent representations, where feature interactions are implicitly captured through deep representation learning Bengio et al. (2013); Guo et al. (2017). Despite their effectiveness, both methods assume centralized access to the full dataset, and latent approaches further obscure provenance, reducing interpretability—constraints that render them unsuitable for privacy-constrained FL settings.

**Federated Learning (FL)** In FL settings, a central server orchestrates multiple clients by aggregating locally–computed model updates (gradients or weights) over communication rounds, keeping raw data on-device McMahan et al. (2017). Consequently, classical FL research has focused on parameter-level aggregation for prediction tasks, proposing algorithms like FedAvg McMahan et al. (2017), FedProx Li et al. (2020), and SCAFFOLD Karimireddy et al. (2020) to address system and statistical heterogeneity. However, these methods have rarely been extended to feature transformation workflows. Existing FL variants that manipulate features operate only in the latent space—e.g., by mixing, aligning, or augmenting embeddings Yoon et al. (2021); Shin et al. (2020); Rasouli et al. (2020), and fall short of producing explicit, interpretable feature constructions. Adapting centralized AutoFE to FL remains challenging: combinatorial candidate search leads to excessive cross-device evaluations, increasing communication and compute costs, and most pipelines entwine feature transformation with iterative model feedback, making most approaches inefficient or impractical to adapt.

Research on combining FL and AFT remains limited despite advances in both fields. While prior studies have explored privacy-preserving **feature selection** via methods like gradient masking, secure aggregation, or differential privacy Zhang et al. (2023); Cassará et al. (2022); Fu et al. (2023), **feature transformation** remains overlooked due to its combinatorial search space, which amplifies communication and privacy challenges. To our knowledge, FLFE Fang et al. (2020) is the first framework for federated feature transformation(FFT). It relies on manually defined operators, scales poorly with growing candidates, and only filters features rather than synthesizing new ones. Recent work, Fed-IIFE Overman & Klabjan (2024) requires uniform local models across clients and server, restricts its evaluation loop to pairwise interactions, limiting scalability and higher-order dependency modeling. Unlike existing methods built on model-level fusion, FEDFT optimizes at the data level by encoding local feature transformation sequences in a shared latent space. Through an encoder–decoder–evaluator architecture, it compresses and refines transformation knowledge for effective inference, cross-client integration, and scalability. Notably, FEDFT enables bi-directional optimization between local and global spaces, offering a novel and effective approach to FFT.

## 3 PRELIMINARIES

**Feature Transformation Sequence.** Feature transformation enhances the tabular feature space by applying mathematical operations to original features. Given a feature space $[f_1, f_2, \ldots, f_N]$, we define each transformation as a mathematical composition over features and operations. For example, one generated feature is $\left(f_1 + \frac{f_1 - f_2}{f_3} - f_2\right)$, where $f_1$, $f_2$, $f_3$ are original features and $+, -, /$ are operations. We adopt the postfix expression encoding from Wang et al. (2023) to represent the entire feature transformation sequence as $\Upsilon = [\gamma_1, \gamma_2, \ldots, \gamma_M]$, where each $\gamma_i$ is a feature index token or an operation. This sequence guides the construction of a more expressive feature space.

**Problem Statement.** Our objective is to develop a novel federated feature transformation framework that constructs an optimized generalizable feature space by aggregating transformation knowledge from heterogeneous local tabular data without sharing raw data. Formally, for a tabular data prediction task, each client holds its own local data set $\mathcal{D} = \{\mathbf{X}, \mathbf{y}\}$, where $\mathbf{X}$ is the feature set and $\mathbf{y}$ is the predictive target. Each client initially generates a set of feature transformation records, denoted as $\mathcal{R} = \{\Upsilon, \mathbf{v}\}$, where $\Upsilon$ represents the collection of transformation sequences and $\mathbf{v}$ denotes the corresponding predictive performances. The records of different clients are uploaded to a central server for subsequent knowledge aggregation. On the server, an encoder $\phi$, decoder $\psi$, and evaluator $\omega$ are jointly trained to embed the feature transformation knowledge from the collected records

into a continuous space $\mathcal{E}$. After training, the optimal global transformation sequence is identified by performing a gradient-based search in the embedding space $\mathcal{E}$ to maximize the weighted average performance across all clients: $\Upsilon^* = \arg\max_{\Upsilon \in \psi(\mathcal{E})} \sum_{k=1}^{K} w_k v_k$, where $w_k$ denotes the weight of client $k$, and $v_k$ denotes the predicted performance on client $k$ for the feature space generated by $\Upsilon$.

## 4 METHODOLOGY

### 4.1 FRAMEWORK OVERVIEW OF FEDERATED FEATURE TRANSFORMATION

Figure 1 presents the complete FEDFT work-flow. Each client first explores its local tabular dataset using an RL-based collector, generating a set of feature transformation records. These records, together with the corresponding sample metrics, are uploaded to the server. On the server side, records from all clients are merged using a sample-aware weighting scheme to produce an unbiased, population-level estimate for each transformation sequence. The aggregated sequence–performance pairs are then used to train an encoder–decoder–evaluator network, which projects the sequences into a shared latent space while learning to predict their performance. After the model converges, a gradient-guided search is performed within this space to identify the sequence with the highest predicted performance. The optimized sequence is then propagated back to clients, where it will be fused with complementary, non-overlapping local transformation sequences to improve client-specific predictive performance.

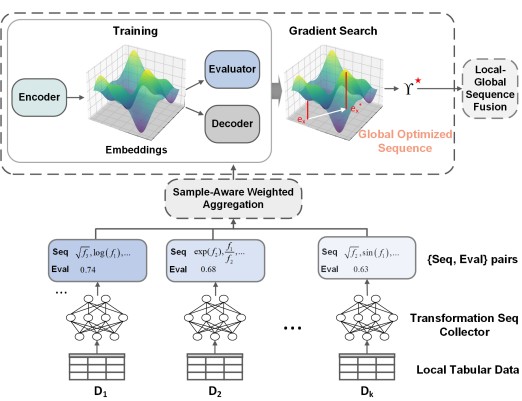

Figure 1: Framework overview of FEDFT. Clients generate transformation records locally and upload them with sample metrics. The server aggregates, learns embeddings, and returns the optimal sequence for local refinement.

#### 4.1.1 AUTOMATED LOCAL FEATURE TRANSFORMATION RECORDS COLLECTION.

To learn a unified and generalizable feature transformation representation, it is essential to thoroughly explore the intrinsic characteristics of tabular data within each client. We apply a reinforcement learning (RL) framework Wang et al. (2022; 2023) on each client's $\mathcal{D}$ to efficiently collect transformation sequence–model performance pairs $\mathcal{R} = \{\Upsilon, \mathbf{v}\}$. Specifically, each client employs a simple three-agent RL framework to collect transformation records. Two agents select candidate features, and a third selects a mathematical operation from a predefined set. The selected features are crossed using the chosen operation to generate a new feature, which is added back to the feature set for further refinement. This process iteratively explores transformation sequences that maximize downstream task performance. *Importantly, the local collector is modular: clients with different resource budgets can plug in non-RL alternatives, and the server aggregates their records in a method-agnostic manner.* The collected transformation sequences and associated model performance scores are subsequently uploaded to the central server for aggregation. As no raw data is exchanged during this process, the framework inherently preserves data privacy and mitigates potential exposure risks.

#### 4.1.2 ENCODER-DECODER-EVALUATOR GLOBAL KNOWLEDGE AGGREGATION.

To derive a unified and generalizable feature transformation sequence, it is essential to aggregate transformation knowledge from diverse clients for further refinement. To this end, we propose an encoder–decoder–evaluator architecture that embeds the collected transformation records from all clients into a shared cross-client embedding space. The objective is to construct a latent space in which each embedding point corresponds to a transformation sequence $\Upsilon$ and its associated predicted model performance $v$, enabling the capture of transferable transformation patterns across heterogeneous tabular data. The model is jointly trained with three objectives: a reconstruction loss ensuring accurate sequence recovery, a performance estimation loss for reliable performance prediction, and a KL regularization term promoting smoothness in the embedding space landscape. This joint objective aligns the embedding space with both the structural and semantic characteristics of transformation sequences, supporting effective global optimization in downstream tasks.

*Encoder $\phi$:* The encoder component effectively maps a given transformation sequence $\Upsilon$ into a continuous embedding vector $\mathbf{E}$. We utilize a single-layer Long Short-Term Memory (LSTM) network Hochreiter & Schmidhuber (1997), where the resulting output is denoted as $\mathbf{E} = \phi(\Upsilon) \in \mathbb{R}^{M \times d}$, with $M$ being the input sequence length and $d$ the embedding hidden dimension.

*Decoder $\psi$:* The decoder component reconstructs the original transformation sequence from the encoder embedding. Implemented as a single-layer LSTM, the decoder takes an initial state $h_0$ and updates its hidden state $h_i^d$ at each step. We apply dot-product attention between the decoder hidden state $h_i^d$ and the encoder outputs to produce a context vector $h_i^e$. The token distribution at step $i$ is

$$P_\psi(\gamma_i \mid \mathbf{E}, \Upsilon_{<i}) = \frac{\exp\left(W_{\gamma_i}(h_i^d \oplus h_i^e)\right)}{\sum_{c \in \mathcal{C}} \exp\left(W_c(h_i^d \oplus h_i^e)\right)}, \text{ where } \mathcal{C} \text{ is the token vocabulary, and } \oplus \text{ denotes concate-}$$

nation. The probability of generating the full sequence is: $P_\psi(\Upsilon \mid \mathbf{E}) = \prod_{i=1}^{M} P_\psi(\gamma_i \mid \mathbf{E}, \Upsilon_{<i})$, which captures the step-wise generation process. We train by minimizing the reconstruction loss $\mathcal{L}_{\text{rec}} = -\log P_\psi(\Upsilon \mid \mathbf{E})$, which encourages assigning high probability to the true sequence.

*Evaluator $\omega$:* The evaluator component estimates the model performance $v$ from the embedding $\mathbf{E}$. We first perform mean pooling operation to obtain a fixed-length vector $\bar{\mathbf{E}} \in \mathbb{R}^d$, which is then passed through a multi-layer feedforward network: $\hat{v} = \omega(\bar{\mathbf{E}})$. The estimation loss is computed as the mean squared error between predicted and real accuracy: $\mathcal{L}_{\text{est}} = \text{MSE}(v, \omega(\bar{\mathbf{E}}))$. To encourage a smooth latent embedding space for the inference stage, we further map $\bar{\mathbf{E}}$ to the mean and log-variance of a Gaussian with MLP layers, obtaining vectors $\boldsymbol{\mu}, \log \boldsymbol{\sigma}^2$. We then construct a KL regularizer with standard reparameterization tricks: $\mathcal{L}_{\text{KL}} = \frac{1}{2} \sum_{j=1}^{d} \left(\mu_j^2 + \sigma_j^2 - 1 - \log \sigma_j^2\right)$.

*Joint Training Loss $\mathcal{L}$:* The encoder, decoder, and evaluator are jointly trained with a weighted loss: $\mathcal{L} = (1 - \lambda)\mathcal{L}_{\text{rec}} + \lambda \mathcal{L}_{\text{est}} + \beta \mathcal{L}_{\text{KL}}$, where $\lambda$ balances predictive performance estimation and $\beta$ regularizes the embedding space to enhance smoothness for exploration. During training, the sequence–performance pairs are embedded into a latent space that is jointly optimized for smoothness and predictive accuracy. Smoothness ensures geometric continuity for effective exploration, while the evaluation loss provides explicit performance guidance. This dual regularization reduces the impact of data heterogeneity and encourages the learning of robust, generalizable representations.

*Gradient-based Optimal sequence Search:* After training, we select the top-$T$ candidate transformation sequences ranked by their average predictive performance $\bar{v}$ across clients. These are encoded into continuous embeddings using the trained encoder, serving as seeds for gradient search. Let $\mathbf{E}$ denote one such embedding. We update $\mathbf{E}$ by ascending along the gradient provided by evaluator $\omega$, following $\widetilde{\mathbf{E}} = \mathbf{E} + \eta \frac{\partial \omega}{\partial \mathbf{E}}$, where $\eta$ is the size of search step. This update is expected to improve predicted performance, i.e., $\omega(\widetilde{\mathbf{E}}) > \omega(\mathbf{E})$. We then decode $\widetilde{\mathbf{E}}$ back to a transformation sequence and select the one with the highest average predictive performance as the globally optimal sequence.

## 4.2 Sample-Aware Weighted Aggregation for Optimization Bias Correction

**Why Sample-Aware Scheme on Global Knowledge Aggregation.** After collecting the transformation–performance records from different clients, we observe that the reliability of these records varies significantly due to heterogeneous local tabular data characteristics. Specifically, varying sample sizes, feature diversity, and numerical stability. This multifaceted heterogeneity is particularly pronounced in tabular settings due to inherent sparsity and heterogeneous feature types. It fundamentally reflects differences in the information content and reliability of local performance signals. Clients with limited data tend to provide less informative and noisier signals, while those with larger, diverse, stable datasets offer more reliable performance indicators. Consequently, directly incorporating these unadjusted scores into the server-side encoder–decoder–evaluator architecture can introduce bias and compromise global model generalizability. To address this, we introduce a sample-aware weighted aggregation strategy, where each client's contribution is scaled by its dataset size, diversity, and stability indicators. This approach enhances statistical stability, reduces the impact of unreliable signals, and supports the learning of a more generalizable embedding space.

As illustrated in Figure 2, even for the same transformation sequence and a fixed downstream model family, the observed performance scores across clients follow distinct distributions. As mentioned earlier, using these unadjusted scores directly would bias the server model and reduce its generalizability. To mitigate this issue and obtain a more reliable model performance estimation, we implement a sample-aware weighted aggregation strategy

that corrects for distributional bias. Specifically, we first collect all transformation records from local clients and construct a non-redundant set of transformation sequences on the server. Each sequence is then broadcast to all $K$ clients, who apply the transformation to their local tabular data and evaluate the resulting model performance. As a result, each transformation sequence is associated with a set of $K$ client-specific performance scores, denoted as $[\hat{v}_1, \hat{v}_2, \ldots, \hat{v}_K]$. We compute client weights by combining two complementary factors—dataset size and data quality—with the overall weight formulated as $w_k = p \cdot w_k^{(size)} + (1-p) \cdot w_k^{(adj)}$. The size-based term $w_k^{(size)} = \frac{|D_k|}{\sum_{j=1}^{K} |D_j|}$ captures the relative sample

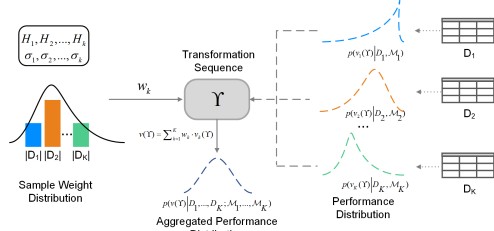

proportion, and the quality-based term $w_k^{(adj)} = \frac{s_k}{\sum_{j=1}^{K} s_j}$, where $s_k = q \cdot H_k + (1-q) \cdot \sigma_k$ incorporates feature entropy $H_k$ (diversity) and stability score $\sigma_k$ (derived from condition number $\kappa_k$).

Figure 2: Sample-Aware weighted aggregation. Sample characteristics and local performance distributions are aggregated for more reliable sequence evaluation.

Here, higher feature entropy indicates more diverse information content, while lower condition numbers (higher stability scores) suggest better numerical stability and reduced multicollinearity risk. This weighting scheme reflects that performance estimates from larger datasets are more reliable, while diverse feature information and numerical stability enhance local signal quality. The final aggregated score for a transformation sequence $\Upsilon$ is computed as: $v = \sum_{k=1}^{K} w_k \cdot \hat{v}_k$.

### 4.3 GLOBAL-TO-LOCAL FEATURE INTEGRATION UNDER MUTUAL INFORMATION CONTROL

**Why Dual Optimization for Local and Global Perspectives Is Essential.** After federated aggregation and server-side optimization, we obtain a globally optimized feature transformation sequence that captures domain-level knowledge across diverse clients. This sequence can serve as a foundation for further domain understanding and analysis. However, clients are typically more concerned with improving performance on their own local predictive tasks rather than contributing to global understanding. Relying solely on local data and patterns may lead to overfitting and hinder generalization, while indiscriminately aggregating local contributions may fail to satisfy individual clients' objectives, threatening the sustainability of collaboration. To address this issue, we adopt a dual-benefit strategy: the globally optimized transformation sequence is fed back to each client, enabling local model refinement while preserving the value of global knowl-

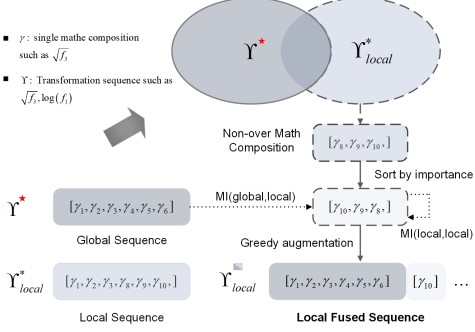

Figure 3: Local-Global sequence fusion. Local feature space is refined by fusing with the global sequence

edge. This not only incentivizes client participation but also ensures that the collaboration remains mutually beneficial. By integrating globally informed transformations, clients can enhance their models in a way that balances domain-level generalization with client-specific adaptability.

As shown in Figure 3, after completing gradient-based search, the server broadcasts the global transformation sequence $\Upsilon^*$ to all clients. To integrate domain-level knowledge with local adaptation, each client augments $\Upsilon^*$ with a small set of locally distinctive features. Each client begins by selecting its local best transformation sequence and extracts derived features as candidates. For each candidate, two criteria are computed: (1) a model-dependent feature importance score for initial ranking, and (2) its redundancy with the global features and the currently selected local ones, measured by mutual information (MI). Let $\mathcal{G}$ denote global features and $\mathcal{C}$ the pool of candidate local features. The client incrementally builds a supplement set $\mathcal{F}$ by greedily selecting the least redundant feature $f \in \mathcal{C}$ at each step, according to the following score: $\text{Score}(f) = \eta \cdot \text{AvgMI}(f, \mathcal{F}) + \delta \cdot \text{AvgMI}(f, \mathcal{G})$, where $\text{AvgMI}(f, .)$ denotes the average mutual information between $f$ and the features in the corresponding set. The process continues until a fixed budget is reached or no candidates remain.

Finally, the client builds its personalized feature set by combining global and selected local features: $\mathcal{Z}_k = \mathcal{G} \cup \mathcal{F}$. where $\mathcal{Z}_k$ is the final augmented feature set used by client $k$ for downstream predictive modeling. This fused representation preserves domain-level knowledge while incorporating client-specific insights, yielding consistent improvements under heterogeneous tabular distributions.

## 5 EXPERIMENT

### 5.1 EXPERIMENT SETTINGS

**Dataset and Evaluation Metrics.** We use 13 publicly available datasets from UCI Kelly et al. (2023), LibSVM Lin (2022), and OpenML Public (2022) to conduct our experiments. Table 4 shows the statistics of the data. Regression tasks were evaluated using the following metrics: 1-Relative Absolute Error (1-RAE), 1-Mean Absolute Error (1-MAE), 1-Root Mean Squared Error (1-RMSE) and coefficient of determination ($R^2$). For classification tasks, assessment was conducted using Precision, Recall, F1-score and AUC-score. The formulae for the F1-score and 1-RAE are given by: F1$= 2 \cdot \frac{\text{Precision}\cdot\text{Recall}}{\text{Precision}+\text{Recall}}$ and 1-RAE $= 1 - \frac{\sum_{i=1}^{n} |y_i - \tilde{y}_i|}{\sum_{i=1}^{n} |y_i - \bar{y}_i|}$, where $y_i$, $\tilde{y}_i$, and $\bar{y}_i$ represent the ground truth, predictions, and the mean of the ground truth, respectively.

**Baseline Methods.** To sufficiently evaluate FEDFT, we consider three baseline methods. First, we compare three transformation sequences: $\Upsilon_I$, an ideal upper bound obtained by directly optimizing on global data; $\Upsilon^*$, the global sequence generated by FEDFT through federated aggregation; and $\Upsilon_{\text{local}}$, the best locally generated sequence on each client. These are evaluated on both a global test set and local client test sets to examine generalization and client-level effectiveness. Second, we integrate FEDFT into standard federated learning frameworks such as FedAvg by allowing each client to maintain its own transformation module, enabling fair comparison with traditional model-weight averaging strategies. Third, we conduct ablation studies to isolate the impact of two key components: FEDFT$^{-u}$ removes the encoder–decoder–evaluator structure, and FEDFT$^{-c}$ removes the sample-aware weighted aggregation. These comparisons collectively show the contribution of global optimization, local adaptation, and aggregation correction to the overall performance of FEDFT.

### 5.2 PERFORMANCE EVALUATION

**Overall Performance.** This experiment aims to answer the following questions: *Can our FEDFT framework effectively capture the domain knowledge of an inaccessible global distribution and enhance local models through globally coordinated feature transformations?* Table 1 shows results in terms of F1 and 1-RAE. **Global** column reports performance on the unseen global dataset for reference, while **Local** column shows the average test performance across clients. We report three variants: $\Upsilon_I$: the upper bound achieved by gradient search with direct access to global data; $\Upsilon^*$: the best sequence selected without global data, based on weighted local performances; $\Upsilon_{\text{local}}$: the best locally generated feature transformation sequences of each client for average performance. We observe that FEDFT consistently produces transformation sequences that outperform locally generated ones in both global and local evaluations. The underlying driver is the complementary effect of global knowledge aggregation and local feature fusion. For global performance, FEDFT aggregates information from all clients, capturing the shared structure of the global data distribution. For local performance, the feature fusion mechanism enriches the local feature space by combining global knowledge with client-specific features, improving local model effectiveness. In summary, FEDFT effectively captures global distribution information while refining global and local feature spaces.

**Comparison with Federated Baselines.** This experiment aims to answer the following questions: *Can our FEDFT framework match or outperform the well ackowledged federated learning methods?* To answer this question, we compare FEDFT with recent feature-transformation–focused FL method Fed-IIFE Overman & Klabjan (2024) and with established federated learning baselines(FedAvg McMahan et al. (2017), FedProx Li et al. (2020), MOON Li et al. (2021) and Fed-NTD Lee et al. (2022)) that follow a different aggregation strategy: while FEDFT shares feature transformation-performance pairs, traditional methods rely on model weight aggregation, requiring all clients to adopt the same model structure as the global model. These methods also incur extra costs, as they maintain both the FEDFT transformation pipeline and a local encoder-decoder-evaluator module. As shown in Table 2, we conduct experiments on three datasets: Openml_586, Wine Quality Red, and Pima Indian datasets, using 1-RAE for regression and F1 score for classification tasks. For each method, we apply the best feature transformation sequence to both global and local partitions, reporting mean and standard deviation across all clients for the latter. We observe

Table 1: Overall performance comparison. Global and Local columns report results on the global and local test sets for three sequence variants: $\Upsilon_I$ (ideal upper bound with global data access), $\Upsilon^*$ (optimal sequence via sample-aware aggregation), and $\Upsilon_{\text{local}}$ (client-wise best local sequences).

| Dataset | Global | | | Local | |
|---|---|---|---|---|---|
| | $\Upsilon_I$ | $\Upsilon^*$ | $\Upsilon_{local}$ | $\Upsilon^*$ | $\Upsilon_{local}$ |
| SpectF | 0.8234 | 0.8164 | $0.7740\pm_{0.0210}$ | 0.8817 | $0.8742\pm_{0.0050}$ |
| Pima Indian | 0.7725 | 0.7669 | $0.7455\pm_{0.0083}$ | 0.7532 | $0.7316\pm_{0.0042}$ |
| SVMGuide3 | 0.8473 | 0.8473 | $0.8415\pm_{0.0176}$ | 0.8439 | $0.8249\pm_{0.0015}$ |
| Wine Red | 0.6843 | 0.6843 | $0.6671\pm_{0.0061}$ | 0.6330 | $0.6283\pm_{0.0089}$ |
| Wine White | 0.6776 | 0.6760 | $0.6729\pm_{0.0028}$ | 0.6646 | $0.6570\pm_{0.0026}$ |
| Housing Boston | 0.6487 | 0.6487 | $0.6431\pm_{0.0039}$ | 0.5826 | $0.5642\pm_{0.0042}$ |
| Airfoil | 0.7892 | 0.7892 | $0.7760\pm_{0.0060}$ | 0.7674 | $0.7529\pm_{0.0022}$ |
| Openml_620 | 0.7168 | 0.7168 | $0.7030\pm_{0.0070}$ | 0.6036 | $0.5935\pm_{0.0011}$ |
| Openml_589 | 0.7421 | 0.7409 | $0.7018\pm_{0.0407}$ | 0.6238 | $0.5627\pm_{0.0571}$ |
| Openml_586 | 0.7742 | 0.7742 | $0.6727\pm_{0.0076}$ | 0.6511 | $0.5198\pm_{0.0119}$ |
| Openml_637 | 0.5702 | 0.5667 | $0.5575\pm_{0.0089}$ | 0.3334 | $0.2884\pm_{0.0370}$ |
| Openml_618 | 0.7457 | 0.7457 | $0.7392\pm_{0.0103}$ | 0.5959 | $0.5819\pm_{0.0256}$ |
| Openml_607 | 0.7395 | 0.7395 | $0.6405\pm_{0.0117}$ | 0.5182 | $0.4367\pm_{0.0218}$ |

$\pm$ indicates standard deviation across clients.

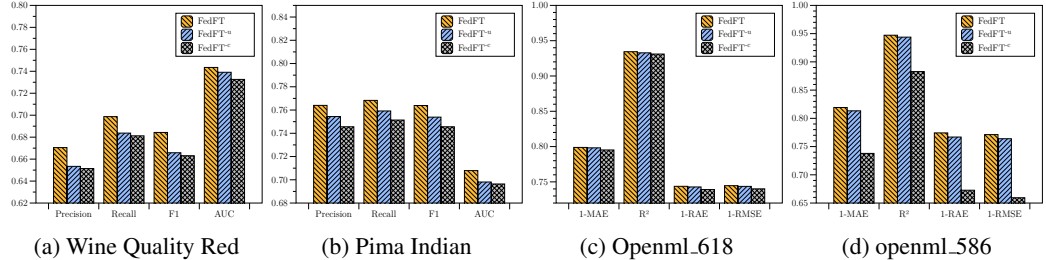

(a) Wine Quality Red     (b) Pima Indian     (c) Openml_618     (d) openml_586

Figure 4: The influence of global aggregation ($\text{FedFT}^{-c}$) and encoder–decoder–evaluator knowledge integration ($\text{FedFT}^{-u}$) in FedFT.

that FEDFT achieves performance comparable to or better than these well-acknowledged models. The underlying reason is that feature transformation sequence-performance pairs carry sufficient information to enable effective aggregation without requiring full model weight sharing. In summary, FEDFT offers an effective and flexible alternative to weight-sharing approaches.

**Ablation Study.** This experiment aims to answer: *Does the sample-aware weighted aggregation and encoder-decoder-evaluator knowledge internalization help to maintain the performance of FEDFT?* To answer this question, we develop two model variants of FEDFT: 1) $\text{FEDFT}^{-u}$, which drops the encoder-decoder-evaluator part, and 2) $\text{FEDFT}^{-c}$, which drops the sample-aware weighted aggregation step in FEDFT. We report comparison results in terms of F1 score or 1-RAE on four datasets: Openml_618, Openml_586, Wine Quality Red and Pima Indian. Figure 4 presents the results on global data. We observe that FedFT significantly outperforms $\text{FedFT}^{-c}$. This improvement is attributed to the fact that local data silos contain only partial and biased data distributions. Without federated aggregation, models trained on these silos perform poorly when generalized to global data. Second, we find that FedFT consistently achieves better performance than $\text{FedFT}^{-u}$. This gain is driven by the encoder-decoder-evaluator module, which not only compresses and smooths the aggregated information but also enables continuous optimization through gradient search, leading to better feature transformation sequences. In summary, both the sample-aware weighted aggregation and the encoder-decoder-evaluator module are essential for preserving the effectiveness of FEDFT and achieving consistent performance improvement over its simplified version.

**Impact of KL-Induced Embedding Smoothness on Latent Feature Space Search.** This experiment aims to answer: *Whether KL divergence loss enhances the smoothness of the refined latent space and improves the gradient-based search.* To analyze this effect, we visualize the latent space embeddings of two models trained on the Pima Indian dataset: one without KL loss and the other

Table 2: Performance Comparison of Different Federated Learning Algorithms

| Method | Openml_586 | | Wine Red | | Pima Indian | |
|---|---|---|---|---|---|---|
| | Global | Local | Global | Local | Global | Local |
| FedAvg | 0.7665 | 0.6413±0.0254 | 0.6706 | **0.6466**±0.0529 | 0.7560 | 0.7513±0.0414 |
| FedProx | 0.6877 | 0.5051±0.0433 | 0.6650 | 0.6413±0.0437 | 0.7457 | 0.7188±0.0244 |
| MOON | 0.7718 | 0.6485±0.0277 | 0.6675 | 0.6440±0.0577 | 0.7552 | 0.7528±0.0623 |
| FedNTD | 0.7661 | 0.6405±0.0257 | 0.6765 | 0.6407±0.0594 | 0.7535 | 0.7556±0.0492 |
| FedCross | 0.7670 | 0.6413±0.0254 | 0.6670 | 0.6392±0.0512 | 0.7540 | 0.7545±0.0557 |
| Fed-IIFE | 0.7538 | 0.6106±0.0186 | 0.6548 | 0.6221±0.1084 | 0.7478 | **0.7578**±0.0354 |
| Ours | **0.7742** | **0.6511**±0.0226 | **0.6843** | 0.6330±0.0087 | **0.7669** | 0.7532±0.0418 |

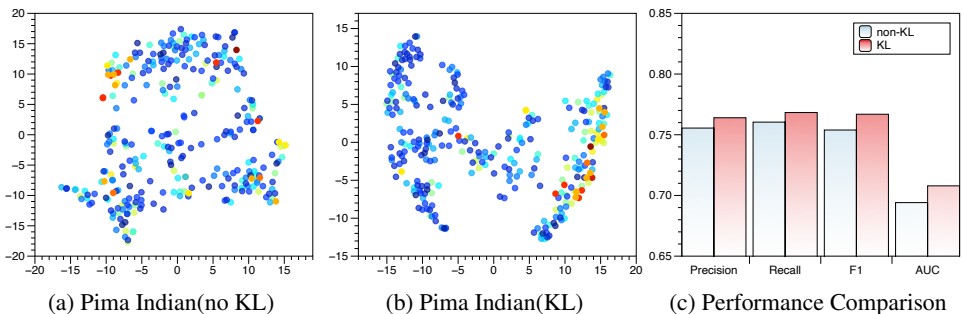

(a) Pima Indian(no KL)    (b) Pima Indian(KL)    (c) Performance Comparison

Figure 5: Comparison of latent space smoothness and downstream performance with and without KL loss. (a)-(b) show smoother and more clustered embeddings with KL loss. (c) shows improved downstream performance after gradient search when KL loss is applied.

with KL loss applied in the Figure 5. Warmer-colored points indicate higher performance. We observe that the model trained without KL loss produces a less smooth latent space, where high-performing sequence embeddings are scattered with large performance variations. Conversely, the model trained with KL loss exhibits a smoother latent space, where high-performing sequence embeddings are more tightly clustered with clear, continuous transitions. This indicates that KL loss encourages smoother structuring of the latent space. We further compare the downstream task performance after applying gradient-based search. Results show model with a smoother latent space enables more effective gradient search, finding higher-quality feature transformation sequences. In summary, KL-induced smoothness enhances latent space structure, improving gradient search.

## 6 CONCLUSION

In this paper, we propose a federated feature transformation framework designed to effectively refine the feature space and aggregate knowledge in tabular data silos settings. The framework centers on two key components: a sample-aware weighted knowledge aggregation module and a local-global sequence fusion scheme. To preserve privacy while enabling knowledge sharing, we introduce a sequence-performance pair communication protocol, allowing clients to share feature transformation knowledge without exposing raw data. To address performance variance and bias from heterogeneous client data, we develop a sample-aware weighted aggregation strategy, which allocates weights to client-reported performance scores based on their local sample characteristics. It ensures that the aggregated evaluation better reflects the overall data distribution and improves server-side training. To dual-optimize both the server and client-side performance, we design a local-global sequence fusion scheme. The server feeds back the globally aggregated transformation sequence, capturing domain-level knowledge, to clients. Each client then fuses this global knowledge with locally distinctive features, improving local predictive tasks while incentivizing client participation. Extensive experiments demonstrate that FEDFT effectively enhances both global and local performance. The proposed sample-aware weighted aggregation mitigates clients' data heterogeneity, while the feedback and fusion scheme benefits individual clients and encourages ongoing contribution and collation across the federation. In future work, we plan to extend FEDFT to dynamic federated environments, where client availability and data distributions evolve over time, and to explore its integration with automated feature generation and task-adaptive personalization strategies.

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

# A APPENDIX

## A.1 EXPERIMENTAL SETTINGS AND REPRODUCIBILITY

### A.1.1 EXPERIMENTAL PLATFORM INFORMATION

All experiments are conducted on an Ubuntu 22.04.5 LTS operating system, powered by an AMD Ryzen Threadripper PRO 7965WX 24-core processor, equipped with two NVIDIA RTX A6000 GPUs (48GB memory each). Python 3.12.7 and PyTorch 2.5.1 are used as the software framework.

### A.1.2 HYPERPARAMETER SETTINGS AND REPRODUCIBILITY

The operation set includes square root, square, cosine, sine, tangent, exponential, cube, logarithm, reciprocal, quantile transformer, min-max scaler, sigmoid, addition, subtraction, multiplication, and division. For data collection, we ran the RL-based data collector on each local client's data for 512 epochs with 10 steps per epoch, gathering a large number of feature transformation–accuracy pairs. For data augmentation, we randomly shuffled each transformation sequence 12 times to increase data diversity and volume, applying a token mask probability of 0.3 and a disorder probability of 0.1. We adopted a single-layer LSTM as both the encoder and decoder backbones, and a three-layer feed-forward network as the predictor. The hidden sizes of the encoder, decoder, and predictor were set to 64, 64, and 200, respectively. The embedding size for both feature ID tokens and operation tokens was set to 32. FedFT was trained with a batch size of 32, a learning rate of 0.001, $\lambda = 0.95$, and $\beta = 0.001$. For inference, we used the top-50 records as seed sequences. For data segmentation, regression datasets were partitioned into equal blocks without random disturbance to preserve segmentation differences. For classification datasets, we used an $\alpha$ ranging from 0.5 to 1.0 to divide the data based on the number of samples in each dataset. For the aggregation mechanism, we allocate 90% weight to performance metrics and 10% to feature diversity and stability considerations. For the baseline experiments presented in Table 2, we adapted our model based on the framework introduced by Zhang et al. Zhang et al. (2025).

## A.2 EXPERIMENTAL RESULTS

### A.2.1 ROBUSTNESS CHECK

This experiment aims to answer: *Does* FEDFT *exhibit robustness when confronted with various machine learning models serving as downstream tasks?* To investigate, we replace the downstream machine learning model with Random Forest (RF), Support Vector Machine (SVM), XGBoost (XGB), K-Nearest Neighbors (KNN), Ridge Regression (Ridge), LASSO Regression (LASSO), and Decision Tree (DT), respectively. Figure 6 presents the comparison results in terms of F1 and 1-RAE on the Ionosphere and Openml_620 datasets. We observe that, regardless of the choice of downstream models, FEDFT consistently facilitates the construction of more generalized feature transformation sequences at a global scale, compared to locally derived sequences. This effect can be attributed to the fact that when local samples are limited or biased, FEDFT aggregates information from multiple clients to capture domain knowledge that reflects the global feature space—knowledge that is otherwise inaccessible to individual clients. Thus, this experiment demonstrates the robustness of FEDFT in refining feature space across different downstream machine learning models.

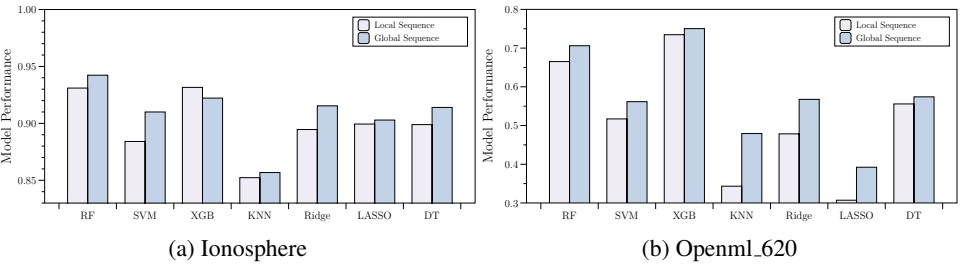

(a) Ionosphere             (b) Openml_620

Figure 6: Robustness Check of FedFT when confronted with different downstream ML models

### A.2.2 IMPACT OF SEED AND STEP HYPERPARAMETERS ON VALIDATION STABILITY AND GRADIENT SEARCH PERFORMANCE

This experiment aims to answer: *How do the seed number and search step size affect the gradient-guided sequence optimization process?* To address this question, we conduct experiments by varying the gradient search step size and the number of top seeds used for optimization. Specifically, we perform evaluations on the Ionosphere and OpenML_586 datasets. For the search step size, we test values ranging from 1 to 20, and for the number of seeds, we evaluate settings from 1 to 50. As shown in figure 7. Our observations show that increasing both the number of seeds and the search step size expands the search space, leading to improved performance. However, regarding the validation rate of newly generated sequences, we find that as the search step increases, the validation rate tends to decline due to greater deviation from the original seed embeddings. Notably, when the initial validation rate is relatively high, this performance drop is less severe.

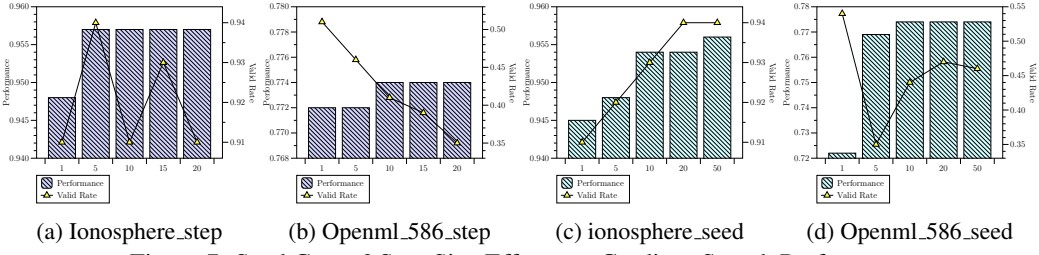

| (a) Ionosphere_step | (b) Openml_586_step | (c) ionosphere_seed | (d) Openml_586_seed |

Figure 7: Seed Count&Step Size Effects on Gradient-Search Performance

### A.2.3 CASE STUDY

We selected the top 10 essential features for prediction from both the globally optimized sequence and a fused sequence on a subset of the Airfoil dataset. We evaluate the traceability and the effect of global-local feature fusion. In Airfoil Global, the chart shows the globally optimized features result, while Airfoil Local presents the fused sequence combining globally optimized features with additional locally critical, non-redundant features. As shown in Figure 8, we observe that the feature spaces differ between global and local data distributions, with some features being uniquely important for specific clients, reflecting data heterogeneity. The fusion operation helps improve local model performance by retaining locally relevant features. The top-ranked features capture key physical interactions: frequency directly relates to acoustic emissions, while combinations such as thickness divided by velocity and frequency multiplied by length reflect how airfoil geometry and flow dynamics jointly influence sound generation. Furthermore, nonlinear transformations, such as sigmoid(thickness), enhance feature expressiveness, enabling domain experts to trace the underlying physical factors and derive interpretable analysis rules for evaluating system performance.

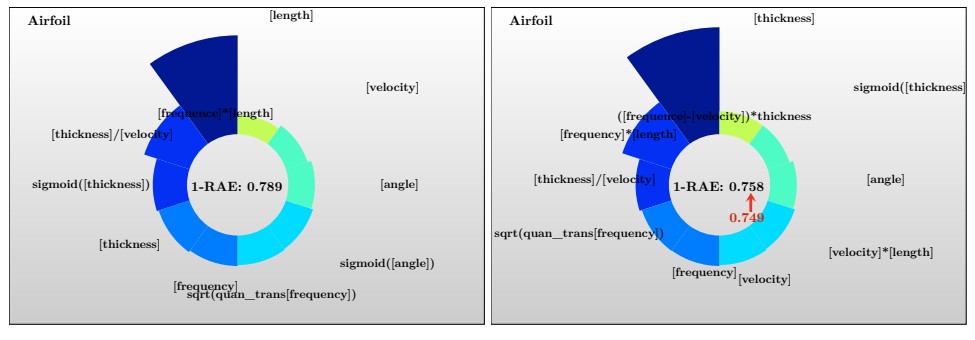

| (a) Airfoil Global | (b) Airfoil Local Fused |

Figure 8: Case study on Airfoil dataset: Global and Local-Fusion

### A.2.4 PERFORMANCE ANALYSIS UNDER VARYING DATA HETEROGENEITY CONDITIONS

This experiment aims to answer the answer: *Does* FEDFT *maintain robustness and provide consistent benefits under varying levels of data heterogeneity across data silos?* To investigate this, we evaluate FEDFT, on three datasets (Wine Red, Spectf, and SVMGuide3) under three distinct data

Table 3: Performance of FEDFT under different heterogeneity levels. We report global sequence performance and local best sequence performance (mean ± std) across three datasets.

| Dataset | Method | Heterogeneity Level | | |
|---|---|---|---|---|
| | | Centralized | Dir($\alpha$=1.0) | Dir($\alpha$=0.5) |
| Wine Red | Global (FEDFT) | 0.686 | 0.6730 | 0.6843 |
| | Global (Local) | - | $0.6661\pm0.0062$ | $0.6671\pm0.0061$ |
| | Local (FEDFT) | - | 0.6519 | 0.6330 |
| | Local (Local) | - | $0.6379\pm0.0077$ | $0.6283\pm0.0089$ |
| Spectf | Global (FEDFT) | 0.878 | 0.7802 | 0.8164 |
| | Global (Local) | - | $0.7740\pm0.0210$ | $0.7740\pm0.0210$ |
| | Local (FEDFT) | - | 0.8174 | 0.8817 |
| | Local (Local) | - | $0.7972\pm0.0013$ | $0.8742\pm0.0050$ |
| SVMGuide3 | Global (FEDFT) | 0.850 | 0.8379 | 0.8473 |
| | Global (Local) | - | $0.8310\pm0.0025$ | $0.8415\pm0.0176$ |
| | Local (FEDFT) | - | 0.8225 | 0.8439 |
| | Local (Local) | - | $0.8044\pm0.0047$ | $0.8249\pm0.0015$ |

heterogeneity settings: 1) Centralized setting: All data is pooled together (baseline); 2) Moderate heterogeneity: Dirichlet distribution with $\alpha = 1.0$; 3) High heterogeneity: Dirichlet distribution with $\alpha = 0.5$. We report F1 scores as the evaluation metric for transformation sequences generated by FEDFT, and compare them with each local client's best sequences. These metrics are evaluated on both global and local test datasets to comprehensively investigate the benefits and robustness of FEDFT, with mean and standard deviation reported across clients for later. We observe that FEDFT consistently refines both global and local feature spaces across different heterogeneity levels. Specifically, for Wine Red and SVMGuide3 datasets, the federated learning performance closely approximates the centralized training baseline, indicating strong robustness to data heterogeneity. For the Spectf dataset, while FEDFT improves performance compared to individual client baselines, a notable performance gap remains relative to the centralized setting. The underlying driver of these results is that FEDFT effectively aggregates knowledge across distributed clients, enabling performance improvements in federated settings. The model successfully captures and shares beneficial transformation patterns across silos. However, extreme cases like Spectf present unique challenges—with only 267 global samples across 44 features partitioned into 3 highly heterogeneous segments. In such scenarios, the severe data scarcity per client significantly distorts local performance signals, limiting the effectiveness of knowledge aggregation. In summary, FEDFT demonstrates robust performance across varying heterogeneity levels and consistently enhances feature space refinement and knowledge aggregation in federated tabular data environments. The model effectively bridges the performance gap between isolated data silos and centralized training, providing substantial benefits even under challenging heterogeneous conditions.

## A.3   HYPERPARAMETER SENSITIVITY OF $\lambda$

This experiment aims to answer: *Is* FEDFT *sensitive to the choice of $\lambda$ during encoder-decoder-evaluator training?* To analyze this, we vary $\lambda$ while keeping all other architectural and training settings fixed, and report the average precision, recall, and F1 over 5-fold validation. We observe that although the metrics show slight fluctuations across different $\lambda$ values, they remain within a narrow range, indicating that our encoder–decoder–evaluator architecture is robust to the choice of $\lambda$ for balancing prediction and reconstruction losses.

## A.4   COMMUNICATION COSTS AND SECURE COMPUTATIONS

We propose to construct transformation sequence-performance pairs without directly sharing the raw data, so that raw features and labels remain on the clients and are never transmitted to the server. The original raw data resides in $\mathbb{R}^{n \times d}$, where $n$ is the sample size and $d$ is the feature dimension.

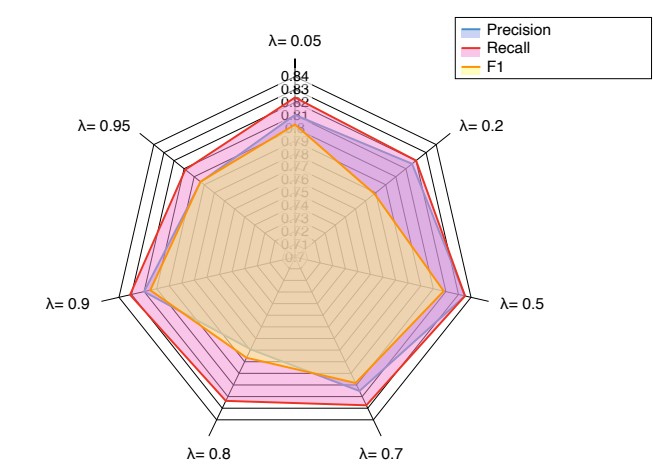

Figure 9: Hyperparameter Sensitivity of $\lambda$ on Spectf

In contrast, our method only requires transmitting the tokenized sequence of length $L$ and a scalar metric, with a payload size of $\mathbb{R}^{L \times 1}$. We assume $n \gg L$. Consequently, communicating abstract sequences significantly reduces communication costs when $n$ is large enough. Furthermore, in scenarios where each client is strictly required not to share explicit local performance statistics or the source of transformation sequences, our architecture can incorporate Secure Aggregation Bonawitz et al. (2017) and the Shuffle Model Erlingsson et al. (2019). This integration allows the server to collectively compute the global ranking and obtain a pool of transformation sequences without accessing individual performance values or tracing the sequence origin, decoupling the data utility from client identity. Moreover, regarding the server-to-client feedback loop, if local clients are restricted from inferring the attribute values or data distributions of other peers through the broadcasted global sequences, our framework supports a Source Obfuscation strategy via Synthetic Sequence Injection. By strategically injecting server-generated "decoy" sequences into the broadcast pool ($\mathcal{S}_{\text{broadcast}} = \mathcal{S}_{\text{real}} \cup \mathcal{S}_{\text{synthetic}}$), we construct a probabilistic barrier. This mechanism leverages the principle of plausible deniability Bindschaedler & Shokri (2016), preventing malicious clients from confidently distinguishing whether a sequence represents a high-value insight from other peers or synthetic noise. It is worth noting that some secure computation techniques can introduce significant computation overhead. To further enhance privacy protection and computational efficiency, it would be beneficial to further improve our proposed method, and we leave it for future exploration.

### A.5 LIMITATIONS

While our current framework employs an LSTM-based sequence encoder, we recognize the potential for further improvements in efficiency and representation capacity through more advanced seq2seq models such as transformers. In future work, we plan to explore transformer-based and other state-of-the-art sequence modeling techniques to enhance computational performance and capture richer feature interaction patterns. Additionally, we aim to further investigate non-IID mitigation strategies tailored to our setting and expand our framework to support broader data types beyond tabular data.

### A.6 DATASET STATISTICS

### A.7 FEATURE ENTROPY AND CONDITION NUMBER

The feature entropy $H$ measures the diversity of features in the dataset. For each feature $j$ in dataset $\mathcal{D}$ with $n$ features:

$$H_j = \begin{cases} -\sum_{i=1}^{u_j} p_{ij} \log(p_{ij}), & \text{if } u_j \leq 10 \text{ (discrete)} \\ -\sum_{b=1}^{B} p_{bj} \log(p_{bj}), & \text{if } u_j > 10 \text{ (continuous)} \end{cases} \tag{1}$$

Table 4: Data Statistics. 'C' for classification and 'R' for regression.

| Dataset | Source | C/R | Samples | Features | Partitions |
|---------|--------|-----|---------|----------|------------|
| SpectF | UCIrvine | C | 267 | 44 | 3 |
| PimaIndian | UCIrvine | C | 768 | 8 | 4 |
| SVMGuide3 | LibSVM | C | 1243 | 21 | 5 |
| Wine Red | UCIrvine | C | 999 | 12 | 5 |
| Wine White | UCIrvine | C | 4900 | 12 | 10 |
| Ionosphere | UCIrvine | C | 351 | 34 | 3 |
| Housing Boston | UCIrvine | R | 506 | 13 | 5 |
| Airfoil | UCIrvine | R | 1503 | 5 | 5 |
| Openml_620 | OpenML | R | 1000 | 25 | 4 |
| Openml_589 | OpenML | R | 1000 | 25 | 5 |
| Openml_586 | OpenML | R | 1000 | 25 | 4 |
| Openml_637 | OpenML | R | 500 | 50 | 5 |
| Openml_618 | OpenML | R | 1000 | 50 | 5 |
| Openml_607 | OpenML | R | 1000 | 50 | 5 |

where $u_j$ is the number of unique values for feature $j$, $p_{ij}$ is the probability of value $i$ in feature $j$, $B$ is the number of bins (set to 10), and $p_{bj}$ is the probability mass in bin $b$. The normalized entropy for each feature is:

$$\hat{H}_j = \frac{H_j}{H_{\max,j}}, \quad \text{where } H_{\max,j} = \begin{cases} \log(u_j), & \text{if discrete} \\ \log(B), & \text{if continuous} \end{cases} \tag{2}$$

The overall feature entropy for the dataset is: $H(\mathcal{D}) = \frac{1}{n} \sum_{j=1}^{n} \hat{H}_j$ The numerical stability is assessed through the condition number of the standardized correlation matrix. Let $\mathbf{X} \in \mathbb{R}^{m \times n}$ be the standardized feature matrix. The correlation matrix is: $\mathbf{R} = \frac{1}{m-1} \mathbf{X}^T \mathbf{X}$ , The condition number $\kappa$ is computed from the eigenvalues $\{a_1, a_2, \ldots, a_n\}$ of $\mathbf{R}$:

$$\kappa(\mathbf{R}) = \frac{a_{\max}}{a_{\min}}, \quad \text{where } a_i > 10^{-10} \tag{3}$$

The stability score $\sigma_k$ is then derived using an exponential decay function: $\sigma_k = \exp(-0.1 \cdot (\kappa - 1))$ This score is clipped to $[0, 1]$, where values closer to 1 indicate better numerical stability.

---

**Algorithm 1:** Postfix transformation sequence conversion

---

**input:** Feature transformation sequence $\Gamma$
**output:** Postfix-notation based transformation sequence $\Upsilon$
$\Upsilon \leftarrow \emptyset$;
**foreach** $\tau \in \Gamma$ **do**
    $S_1, S_2 \leftarrow \emptyset, \emptyset$;
    **foreach** *token $\gamma$ in $\tau$* **do**
        **if** *$\gamma$ is a left bracket* **then**
            $S_1$.push($\gamma$);
        **else if** *$\gamma$ is a right bracket* **then**
            **while** *$(t \leftarrow S_1.pop())$ is not a left bracket* **do**
                $S_2$.push($t$);
        **else if** *$\gamma$ is an operation* **then**
            **while** *$S_1.peek()$ is not a left bracket* **do**
                $S_2$.push($S_1$.pop());
            $S_1$.push($\gamma$);
        **else**
            $S_2$.push($\gamma$);
    **while** $S_2 \neq \emptyset$ **do**
        $\Upsilon$.append($S_2$.pop(0));
    **if** *$\tau$ is not the last element in $\Gamma$* **then**
        $\Upsilon$.append($\langle$SEP$\rangle$);
prepend $\langle$SOS$\rangle$ and append $\langle$EOS$\rangle$ to $\Upsilon$;
**return** $\Upsilon$;

---

## A.8 POSTFIX TRANSFORMATION CONVERSION

Algorithm 1 converts each feature-transformation sequence $\Gamma$ written in infix form into a single postfix-notation sequence $\Upsilon$. We maintain a list $\Upsilon$ and two stacks $S_1$ and $S_2$. For every transformation expression $\tau \in \Gamma$, we scan its tokens from left to right: feature ID tokens are pushed onto $S_2$, left parentheses are pushed onto $S_1$, operators trigger a pop from $S_1$ to $S_2$ until the top of $S_1$ is a left parenthesis, and right parentheses trigger a pop from $S_1$ to $S_2$ until the matching left parenthesis, which is discarded. After finishing one expression, we flush $S_2$ into $\Upsilon$. Between two expressions we insert a special token <SEP>, and we finally wrap the whole sequence with <SOS> at the beginning and <EOS> at the end.

