# OpenReview forum: "Federated Feature Transformation with Sample-Aware Calibration and Local–Global Sequence Fusion"
_ICLR.cc/2026/Conference — Submitted to ICLR 2026_

### Official Review · Reviewer_YYNX · 2025-10-26

**Soundness:** 2
**Presentation:** 3
**Contribution:** 2
**Rating:** 2
**Confidence:** 5

**Summary:**

This paper proposes a framework named FEDFT for federated feature transformation (FFT) on tabular data. Instead of sharing raw data or model gradients, clients generate feature transformation sequences and associated performance scores, which are uploaded to a central server. The server aggregates these records via a sample-aware weighted scheme, trains an encoder–decoder–evaluator network to embed sequences into a shared latent space, and performs gradient-based optimization to explore better global transformation sequences. The globally optimized transformation is then fused with local distinctive features to improve local performance. Experiments on 13 tabular datasets demonstrate improvements over local transformation baselines and some FL methods.

**Strengths:**

1. This paper introduces a joint embedding–reconstruction–performance estimation design, plus local–global fusion, showing architectural completeness.

2. This paper moves beyond traditional parameter aggregation and explores federated feature transformation, which is underexplored but practically relevant for tabular settings.

**Weaknesses:**

1. Although raw data is not shared, transmitting performance values can leak distributional statistics. No comparison with established privacy-preserving FL techniques such as differential privacy or secure aggregation. The paper itself acknowledges lack of formal guarantees.

2. The transformation sequences are symbolic and non-differentiable. Updating continuous embeddings and decoding them does not guarantee validity or monotonic improvement. Valid decode rate is low (Figure 7), and the optimization might be unstable or ill-defined.

3. Works like FLFE and Fed-IIFE are cited but not compared experimentally, limiting the strength of the claimed contribution.

4. Broadcasting sequences to all clients for re-evaluation can be expensive, contradicting efficiency motivations.

5. Some methodological descriptions lack clarity. For example, the KL term is referenced but not precisely defined, hyperparameters (e.g., p, q in weighting) appear heuristic without sensitivity studies, potential leakage in using aggregated validation performance for server optimization

6. The introduction highlights three major challenges that this work aims to address. However, the experiments do not fully demonstrate that the proposed method effectively resolves all of these challenges. As a result, the underlying working mechanism remains unclear, and the claimed contributions are significantly weakened.

**Questions:**

Does the method truly provide privacy-preserving global feature transformation?

Does the proposed sample-aware weighted aggregation truly reduce bias under non-IID data?

What is the causal mechanism by which global-to-local fusion brings improvements?

---

> ### Author Response · Authors · 2025-11-22
>
> Thank you for your insightful advice.
>
> **W1/Q1.**
>
> Our privacy setting follows the horizontal Fed-IIFE–style: clients never share raw features, labels, or local model parameters, only records from which the global sequence is learned. Mechanisms such as secure aggregation, shuffle-model protocols, or source-obfuscation schemes can be incorporated if stronger security guarantees are required. We have added a discussion in the revision.
>
> **W2.**
>
> We agree that transformation sequences are symbolic and non-differentiable; our method does not backpropagate through discrete operators. As in AutoFE and NAS, our problem is a combinatorial search over transformation sequences, not a convex parameter optimization task, so step-wise monotonic improvement guarantees are generally not available even in centralized AutoFE. Progress comes from guided exploration plus selection, rather than every update strictly decreasing a loss.
>
> Validity is enforced by the decoder and simple filtering. The decoder is grammar-constrained (operator/arity/type rules), and any remaining syntactically or type-invalid sequences are discarded and never evaluated or used for training. As shown in Figure 7, the valid-decode rate is high on Ionosphere (>0.91 for all step sizes) and stays in a 0.35–0.55 range on Openml-586, where performance still improves; in practice, we choose the step size and the number of sampled candidates to stay in such a stable regime.
>
> The optimization itself is well-defined: the encoder–decoder–evaluator is trained on valid decoded sequences only, using a smooth surrogate loss in the embedding space, and the server maintains a pool of evaluated sequences and keeps only the best-performing ones after each gradient-guided sampling round. This is the same paradigm used in encoder–decoder based neural combinatorial optimization and AutoFE: the continuous model shapes a proposal distribution, and empirical monotonic improvement comes from selection over evaluated discrete candidates, not from per-step guarantees.
>
> **W3.**
>
> We have included additional experiments comparing FedFT with Fed-IIFE. Note that FLFE is excluded due to code/data unavailability, as also noted in the Fed-IIFE paper. We observe that FedFT maintains better performance.
>
> | Dataset | Metric | Ours (FedFT) | Fed-IIFE |
> | :--- | :--- | :--- | :--- |
> | **Openml_586** | Global| **0.7742** | 0.7538 |
> | | Local| **0.6511 ± 0.0226** | 0.6106 ± 0.0186 |
> | **Wine Red** | Global| **0.6843** | 0.6548 |
> | | Local| **0.6330 ± 0.0087** | 0.6221 ± 0.1084 |
> | **Pima Indian** | Global| **0.7669** | 0.7478 |
> | | Local| 0.7532 ± 0.0418 | **0.7578 ± 0.0354** |
>
> **W4.**
>
> Thank you for pointing this out. Evaluating candidate feature transformations is inherent to automated feature engineering: both centralized AutoFE and recent federated AutoFE methods (e.g., Fed-IIFE) must evaluate many candidates, and Fed-IIFE explicitly notes that naively embedding such pipelines into FedAvg-style FL is infeasible because of the large number of federated evaluations required.
>
> In our framework, each client runs its chosen AutoFE routine under a fixed budget and uploads only a bounded set of transformation–performance records. The server trains the encoder–decoder–evaluator on these records and selects a small number of global sequences (capped in our experiments, e.g., within 2000), so the set of sequences that are ever broadcast and re-evaluated is controlled by design.
>
> Broadcasting these sequences triggers a single, budgeted round of local re-evaluation with each client’s own tabular model, rather than many rounds of federated training over all candidates. This sequence-based aggregation is our efficiency mechanism: we avoid multi-round FL over a large candidate pool and instead perform one global update and one alignment step on a controlled number of sequences.
>
> **W5/Q1.**
>
> (1)We have added the KL-term notation in the revision. (2) We added a hyperparameter λ in Appendix A.3 and observe that FedFT is robust to its value. (3) If this is required, we can incorporate the techniques mentioned in answer to W1/Q1.
>
> **W6.**
>
> We believe the above clarification and additional empirical results address this concern.
>
> **Q2.**
>
> We follow the same sample-aware weighting philosophy as Fed-IIFE and FedAvg, and Better performance on global data and Appendix A.2.4 demonstrates that this choice is empirically validated.
>
> **Q3.**
>
> The effect comes from how the fused feature set reshapes the information each client sees. The global sequence is learned from transformation–performance signals aggregated over all clients, so the induced features capture stable cross-client regularities and act as a low-variance backbone. Each client then adds a small set of locally selected features that are explicitly important and non-redundant w.r.t. the global ones, yielding a feature set that empirically improves predictive power and reduces overfitting compared to purely local or purely global transformations.

---

### Official Review · Reviewer_RWz2 · 2025-10-30

**Soundness:** 2
**Presentation:** 2
**Contribution:** 2
**Rating:** 4
**Confidence:** 4

**Summary:**

This paper proposes a novel federated feature transformation framework designed to construct an optimized and generalizable feature space without sharing raw data.
Each client locally generates feature transformation sequences  and their corresponding predictive performances . These records are uploaded to a central server, where an encoder–decoder–evaluator network jointly learns to embed transformation knowledge into a continuous space .
A gradient-based search in this embedding space identifies the globally optimal transformation sequence that maximizes the aggregated weighted performance across all clients.
The framework aims to improve generalization under heterogeneous data distributions by transferring transformation knowledge rather than raw features or gradients.

**Strengths:**

Well-designed framework — The encoder–decoder–evaluator architecture is conceptually sound and allows transformation knowledge to be encoded and optimized collaboratively without exposing local data.
Potential for cross-domain generalization — By learning transformation patterns rather than model weights, the framework could generalize better across heterogeneous or non-IID clients.
Privacy-preserving design — The method respects the federated setting and does not transmit raw data, which aligns with privacy-by-design principles.

**Weaknesses:**

(1) High computational complexity of the RL search process. The optimization of feature transformation sequences through RL or gradient-based exploration in embedding space is likely to be time-consuming. The paper lacks an explicit analysis of the computational and communication costs, which raises concerns about scalability to high-dimensional tabular datasets or large numbers of clients.
(2) Lack of theoretical analysis. Lack of convergence or optimality guarantees for the global optimal sequence search.
(3) Limited comparison with recent federated learning algorithms. The experiments mostly benchmark against older FL baselines (e.g., FedAvg, FedProx). Moreover, since the method is related to feature engineering, the comparisons do not include papers related to feature engineering.
(4) Reinforcement learning design is underexplained.The paper mentions “postfix encoding” and search but does not provide sufficient detail about the RL reward structure, policy updates, or exploration-exploitation balance.

**Questions:**

Could the authors provide complexity analysis for both local generation of records and server-side embedding training?
Please add comparisons regarding the latest feature engineering related work in federated learning.
Please provide a detailed explanation of the training process in the reinforcement learning part.
Add proofs for the convergence or optimality guarantees of the global optimal sequence search.

---

> ### Author Response · Authors · 2025-11-22
>
> Thank you for the valuable question.
>
>
> **W1/Q1.**
>
> **The RL-based search is only one possible local generator; any cheaper AutoFE or heuristic method can be used instead, which is why our framework is model-agnostic on the client side.** In practice, each client can run its chosen AutoFE routine with a fixed search budget and a cap on the number of transformation–performance records it outputs, so the cost is comparable to running a standard centralized AutoFE method on that client’s data and does not add extra federated overhead.
> **On the server, we only train an encoder–decoder–evaluator on the uploaded records; it sees a limited number of sequences with a fixed embedding size, so the training time is small and does not grow with the original feature dimension**. Communication is one-shot: we require one round where each client uploads a bounded list of records, and one round where the server broadcasts the global sequence; the subsequent global-to-local alignment is done locally based on sample weights and does not incur extra communication. Unlike FedFT, the recent federated feature-transformation method Fed-IIFE applies FedAvg-style aggregation of mutual information for every pair of features, which becomes impractical as either the feature dimension or the number of clients grows.
>
>
>
> **W2.**
>
> Our problem is a combinatorial feature-transformation search, similar to AutoFE and neural architecture search, where the space of discrete sequences is highly non-convex and grows super-exponentially with sequence length, so global optimality or convergence to the global optimum is not attainable even in the centralized setting. Existing AutoFE and federated AutoFE methods likewise rely on heuristic or RL-style search and, to our knowledge, do not provide global optimality guarantees, so our focus in this work is on the algorithmic design and empirical gains rather than new theory for this class of problems.
>
> **W3.**
>
> We have extended our experiments to include both a recent federated AutoFE method (Fed-IIFE) and a 2024 FL optimizer (FedCross). Under the same tabular setting, it shows that FedFT matches or improves the performance of both methods.
>
> **As noted in the Fed-IIFE paper, many centralized AutoFE pipelines are hard to or can’t extend to FL because they expand a large number of engineered features and rely on operations that are not FL-friendly** (e.g., LightGBM evaluations in OpenFE), and FLFE is not reproducible due to non-public data and code.
>
> **Our design (collecting and aligning transformation–performance records) allows us to run traditional FL methods while keeping clients model-agnostic; for this reason**, we retain FedAvg, FedProx, MOON, and FedNTD as strong standard baselines, and use Fed-IIFE and FedCross as recent checks that our method remains competitive with both federated feature-engineering approaches and newer FL optimizers.
>
> | Dataset | Metric | Ours (FedFT) | FedCross |
> | :--- | :--- | :--- | :--- |
> | **Openml_586** | Global| **0.7742** | 0.7670 |
> | | Local | **0.6511 ± 0.0226** | 0.6413 ± 0.0254 |
> | **Wine Red** | Global| **0.6843** | 0.6670 |
> | | Local | 0.6330 ± 0.0087 | **0.6392 ± 0.0512** |
> | **Pima Indian** | Global| **0.7669** | 0.7540 |
> | | Local| 0.7532 ± 0.0418 | **0.7545 ± 0.0557** |
>
> | Dataset | Metric | Ours (FedFT) | Fed-IIFE |
> | :--- | :--- | :--- | :--- |
> | **Openml_586** | Global| **0.7742** | 0.7538 |
> | | Local| **0.6511 ± 0.0226** | 0.6106 ± 0.0186 |
> | **Wine Red** | Global| **0.6843** | 0.6548 |
> | | Local| **0.6330 ± 0.0087** | 0.6221 ± 0.1084 |
> | **Pima Indian** | Global| **0.7669** | 0.7478 |
> | | Local| 0.7532 ± 0.0418 | **0.7578 ± 0.0354** |
>
> **W4/Q1.**
>
> As our contribution lies in innovatively adapting these unique designs to resolve challenges of extending centralized feature transformation to federated learning settings, we chose to cite these works instead of repeating their technical details in the main text. In the revision, we have added a short appendix subsection for better readability.
>
> Wang, D., Xiao, M., Wu, M., Zhou, Y., & Fu, Y. (2023). Reinforcement-enhanced autoregressive feature transformation: Gradient-steered search in continuous space for postfix expressions. Advances in Neural Information Processing Systems, 36, 43563-43578.
>
> Wang, D., Fu, Y., Liu, K., Li, X., & Solihin, Y. (2022, August). Group-wise reinforcement feature generation for optimal and explainable representation space reconstruction. In Proceedings of the 28th ACM SIGKDD Conference on Knowledge Discovery and Data Mining (pp. 1826-1834).

---

### Official Review · Reviewer_2gHq · 2025-11-04

**Soundness:** 2
**Presentation:** 2
**Contribution:** 2
**Rating:** 2
**Confidence:** 3

**Summary:**

This paper proposes federated feature transformation for tabular data. The authors point out that both federated learning and feature transformation have been investigated but their intersection has relatively been unexplored. To alleviate the data heterogeneity issue among various clients, the proposed FedFT employs sample-aware weighting strategy and global-to-local feature integration. Experiments show that FedFT improves performance over the selected baselines.

**Strengths:**

1. Sample-aware weighting that considers both data size and quality seems effective.
2. Exploring unique challenges for tabular data can be interesting.
3. Federated feature transformation is relatively unexplored area.

**Weaknesses:**

- FL has largely been explored for image and text benchmarks but this paper emphasizes tabular data. What is the unique challenge of tabular data compared to other data, such as images and texts? Why not reusing existing techniques for image and text data? Why is the proposal not applied to other data formats?

- Contributing to local clients via federated learning has been also investigated in personalized federated learning (PFL). How is the proposal differentiated from PFL and what is the performance gap?

- FL typically trains a deep learning model itself in a federated manner. In my understanding feature transformation is already included in the model since it extracts deep representations for each input data. What is the unique value of federated feature transformation compared to federated model training? Conceptual illustration comparing feature transformation and other directions can be helpful.

- References are old considering exploding publications on ML these days, which means the research topic may not be important or outdated. Please add more recent papers and if relevant papers do not exist, the authors should justify why this field has been overlooked.

- Similarly, the FL baselines are old; FedNTD published in 2022 is the latest method in the paper. Baselines should include more recent and advanced methods published in 2024 and 2025.

**Questions:**

Please see the weaknesses.

---

> ### Author Response · Authors · 2025-11-22
>
> Thank you for the valuable question.
>
> **W1.**
>
> (1) Our setting is federated tabular feature transformation, rather than federated deep representation learning. Clients apply explicit operators (log, ratio, cross, etc.) to named columns to build symbolic features that are traceable and reusable with standard tabular models, as is common in applications like credit scoring and risk modeling. This is structurally different from image/text FL benchmarks, where deep networks learn implicit latent representations over fixed-shaped arrays.
>
> (2) Classical FL methods (FedAvg/FedProx) only aggregate parameters of a fixed neural network and don’t operate on symbolic feature-transformation sequences. Centralized AutoFE methods (TTG/GRFG/OpenFE) search a large transformation space with many train–validate cycles and, as Fed-IIFE notes, are hard to extend to FL because they expand huge engineered feature sets and rely on FL-unfriendly operations such as LightGBM boosting. Our method fills this gap with a sequence-level aggregation layer: local AutoFE is used only to produce compact (sequence, performance) records, and only these are communicated once.
>
> (3) Our mechanism is abstract: clients report transformation–performance records, and the server trains an encoder–decoder–evaluator to propose a global sequence. In this paper, we instantiate it only for tabular data—using a column-wise scalar operator vocabulary, postfix-encoded sequences, and tabular validation models—while applying the idea to images or text would require modality-specific operator vocabularies, local collectors, and evaluation models, which we leave for future work.
>
> **W2.**
>
> PFL methods keep input representations fixed and personalize model parameters on each client. In contrast, we keep prediction models simple and instead learn explicit tabular feature-transformation sequences at the server from aggregated records, then fuse this global sequence with locally selected transformations to form each client’s feature set. Since our method operates at the feature-engineering layer while PFL acts on model parameters, a direct “PFL vs. ours” comparison would require designing a combined system, which is beyond the scope of this work. Instead, we evaluate in a federated tabular AutoFE setting against strong non-personalized FL baselines under identical models, and Section 5 shows that adding our federated feature transformation consistently improves their performance; these transformed features could also serve as inputs to future PFL methods.
>
> **W3.**
>
> Federated deep models indeed learn latent representations as part of end-to-end training. **Our goal is different**: we learn explicit tabular feature-transformation sequences that can be reused as a traceable feature engineering layer for any downstream tabular model. Classical FL optimizes a specific model; the learned representation is tied to that architecture and task. In contrast, our method aggregates cross-client knowledge in the feature space and outputs symbolic transformation sequences that map raw columns to engineered features. Once learned, these sequences can be reused with different tabular learners without re-running FL.
>
> **In many tabular domains such as credit scoring and medicine, individual columns already have clear semantics and theoretical meaning, and practitioners routinely monitor and audit engineered features rather than less clear deep representations**. Our method brings this line of work into the federated setting.
>
> **W4.**
>
> Recent centralized AutoFE for tabular data, including OpenFE (ICML 2023), U-GFT (KDD 2024), LLM-based feature generation with tree reasoning (NIPS 2024), and evolutionary LLM-based feature transformation (AAAI 2025), to make clear that **automated feature generation for tabular data is an active line of research**.
> While FL work has mainly focused on parameter-level training of deep models and most AutoFE methods remain centralized, recent papers such as Federated Automated Feature Engineering (2024–2025) show that **federated tabular AutoFE is only now emerging, not an outdated topic**.
>
> **W5.**
>
> We additionally include the 2024 FL optimizer FedCross experiment, which shows that FedFT remains competitive. Our setting is model-agnostic federated tabular AutoFE: clients may run different AutoFE models and only upload transformation–performance records. In contrast, FedAvg-style methods require the same deep model structure and are hard, even infeasible, to apply to federated feature transformation, as mentioned in answer to weakness 1.
>
> | Dataset | Metric|Ours (FedFT) | FedCross |
> | :--- | :--- | :--- | :--- |
> | **Openml_586** | Global| **0.7742** | 0.7670 |
> | | Local| **0.6511 ± 0.0226** | 0.6413 ± 0.0254 |
> | **Wine Red** | Global| **0.6843** | 0.6670 |
> | | Local| 0.6330 ± 0.0087 | **0.6392 ± 0.0512** |
> | **Pima Indian** | Global| **0.7669** | 0.7540 |
> | | Local| 0.7532 ± 0.0418 | **0.7545 ± 0.0557** |

---

### Meta-Review · Area_Chair_9hTL · 2026-01-06

**Summary:**

This paper proposes a federated feature-transformation framework for tabular data, where clients upload symbolic feature-engineering sequences rather than models or raw data. Although the direction of federated learning for tabular data is interesting and underexplored, the reviewers have the following concerns:

- Unclear distinction from existing personalized FL and federated representation-learning approaches
- Outdated baselines
- Lack of formal privacy guarantees
- scalability
- heuristic symbolic search without theoretical support

**Reviewer Concerns:**

**Addressed in rebuttal**

- Comparisons to more recent methods

- Difference between symbolic feature engineering and deep FL

**Still outstanding**

- Limited novelty in comparison with other FL methods

- Complexity/scalability analysis

- Symbolic search validity and robustness concerns

- Evaluation remains less comprehensive

**Reviewer Scores:**

The reviewers didn't reply to the authors' rebuttal. But I believe it's hard for them to significantly increase their ratings.

---

### Decision · Program_Chairs · 2026-01-26

Reject